# Improving Online Rent-or-Buy Algorithms with Sequential Decision Making and ML Predictions

**Soumya Banerjee**

**Department of Mathematics and Statistics**
Minnesota State University
Mankato, MN 56001
`banerjee.soumyadipta@gmail.com`

## Abstract

In this work we study online rent-buy problems as a sequential decision making problem. We show how one can integrate predictions, typically coming from a machine learning (ML) setup, into this framework. Specifically, we consider the ski-rental problem and the dynamic TCP acknowledgment problem. We present new online algorithms with or without predictions and obtain explicit performance bounds in-terms of the accuracy of the prediction. Our algorithms are close to optimal with accurate predictions while hedging against less accurate predictions.

## 1 Introduction

Online algorithms embody decision making in an environment with uncertainty. By design, such algorithms disregard special features of the input and aim to provide robust performance guarantees on *all* possible inputs. Such algorithms have been successfully used in many real world problems, see [2] for a thorough discussion.

Machine learning (ML) paradigm aims to identify statistically significant features of the input and leverage these features to generate precise future predictions at the cost of increased error in worst case scenario (**cf.** the no free lunch theorem). Typically, ML systems are trained by minimizing some expected loss function and such system may have large error on outliers. ML systems are also sensitive to training and test data and they can perform badly due to adversarial examples or distribution drift.

A natural question is: *Is it possible to enhance an online algorithm with a predictor that leverages good input predictions while hedging against bad predictions?*

It is reasonable to expect that performance is proportional to the exactness of the prediction. But, naively following predictions, even if a good predictor is available, can lead to substandard performance, see [15, §2.1] for an example. It is important to quantitatively relate the performance to the precision of the prediction. This has important practical implications, e.g. performance gains from a ML setup may not justify the the additional cost of implementing it.

The metric that is commonly used to evaluate the performance of an online algorithm is called *competitive ratio* (see [2]). The competitive ratio of an online algorithm $\mathcal{A}$ is the smallest number $c$, where $c \geq 1$, such that the (expected) cost of the (possibly randomized) algorithm is bounded by the expected cost of the optimal *offline* algorithm over all possible inputs

$$\mathsf{E}(\mathsf{Cost}(\mathcal{A})) \leq c\,\mathsf{E}(\mathsf{Cost}_{\mathsf{opt}}) + \alpha, \quad \text{where } \alpha \text{ is a fixed constant.}$$

Recently, several authors have considered the problem of *enhancing* classical online algorithms using ML advisors, see [13, 12, 15]. The general framework used in these studies consist of modifying an online algorithm $\mathcal{A}$ to an online algorithm $\mathcal{A}^{ML}$ such that:

1. No assumptions are made about the implementation details of the predictor.

2. The algorithm $\mathcal{A}^{ML}$ accepts the prediction as an input.

3. A variable $\eta$, roughly measuring the prediction error of the ML advisor, is introduced. (The precise definition of the variable is problem dependent.)

4. The competitive ratio of the algorithm $\mathcal{A}^{ML}$ is calculated as a function of $\eta$. It is denoted by $\mathsf{c}(\eta)$.

Following [12, 15], the algorithm $\mathcal{A}^{ML}$ is called $\gamma$-*robust* if $\gamma$ is the least upper bound of $\mathsf{c}(\eta)$ for all prediction errors $\eta$. If $\mathsf{c}(0) = \beta$, then $\mathcal{A}^{ML}$ is said to be $\beta$-*consistent*. In general, robustness bounds the performance against *all* predictions, and consistency measures the performance in the ideal case of perfect predictions.

In practice, predictions are seldom perfect. In such a situation, consistency and the robustness of an algorithm are not enough to provide a precise estimation of the competitive ratio. Conceptually, the quantity $\eta$ measures the prediction error of a statistical model. Inherently $\eta$ estimates a *random variable*. While consistency of an estimator, in the sense of statistics i.e. $\mathsf{E}(\eta) = 0$, is desirable, it is not the only important factor. The variance of $\eta$ often plays a significant role. For example, assuming $\mathsf{c}(\eta)$ is a sufficiently smooth function of $\eta$ in a neighborhood of zero, consider the Taylor expansion of $\mathsf{c}(\eta)$ around zero: $\mathsf{c}(\eta) = \mathsf{c}(0) + \mathsf{c}'(0)\eta + \mathsf{c}''(0)\eta^2/2 + \dots$. Taking expectation we see that on an average the variance of $\eta$ dominates the value of competitive ratio for imperfect predictions. Robustness and consistency are not sufficient to capture this functional dependence.

We consider a framework where instead of $\eta$ we consider a natural random variable (depending on the problem) such that its tail probability encapsulates the prediction error. We call it the *accuracy* of the predictor. We express the competitive ratio of our algorithms in terms of this parameter. This formulation generalizes the notions of robustness and consistency and provides a precise quantitative dependence of the competitive ratio on the precision of the predictor.

**Our problems**    We consider the ski rental problem and the dynamic TCP acknowledgment problem. These problems are well known instances of rent/buy problems. At each step, the algorithm must decide between *renting* (which incurs small incremental cost) or *buying* (which incurs a large upfront cost) in the face of uncertain future, see §2 and §3 for a detailed description of the problems. For these problems, several randomized algorithms with a competitive ratio of $\frac{e}{e-1}$ are known, see [10, 9, 16, 14, 11].

**Our approach**    Abstractly, one can consider a rent-buy problem as an instance of a sequential decision making problem. At each epoch the *algorithm* chooses a decision from a *decision space* and the *environment* (assumed oblivious) provides a feedback. Every decision incurs a cost and the goal of the *algorithm* is to minimize the total cost. We present a sequential decision theoretic framework for the ski rental problem. The decision theoretic framework for the dynamic TCP acknowledgment problem is due to Seiden (see [16]).

In this setting it is natural to view ML-predictor as expert advice. Algorithms combining expert advice to achieve near optimal performance have been extensively studied under the rubric of *boosting* in machine learning literature, see [5, 17, 3, 1, 11]. Specifically, we use Freund and Schapire's beautiful Hedge($\beta$) algorithm [5]. We note that this algorithm is a specific instance of the more general philosophy of *multiplicative weights update method* which appears as the key idea in several well known algorithms in computer science, see [1] for a detailed discussion.

**Main results**    Let $\beta \in (0, 1)$ be a hyper parameter. For the ski rental problem, we propose a new online algorithms (with or without ML-prediction) that has competitive ratio of $\frac{\log(\beta)}{\beta - 1}$. It turns out that at $\beta = 0.4$, this competitive ratio is lower than $\frac{e}{e-1}$ (see §2.1.1).

In the case of dynamic TCP acknowledgment problem, we obtain a algorithm that has a competitive ratio of $\frac{\log(\beta)}{(\beta - 1)}$ with respect to any ML-predictor. In the terminology of [12], our algorithm is $\frac{\log(\beta)}{\beta - 1}$

consistent. We note that in this case the accuracy of the prediction must be above a certain threshold to achieve a performance that is better than $\frac{e}{e-1}$. We note that our framework generalizes easily to include multiple ML-predictors (see [6] for related work in this direction).

## 2 Ski Rental Problem

At a ski resort, ski equipment rents at \$1 a day and it is priced at \$b. It snows for $\mathbf{T}$ days (where $\mathbf{T}$ is unknown). The ski rental problem asks for a strategy that minimizes the total cost incurred by a skier.

In hindsight, if $\mathbf{T} \geq \mathsf{b}$, then the best strategy is to *buy* the equipment and if $\mathbf{T} < \mathsf{b}$ then the best strategy is to *rent* the equipment. The algorithm: If it snows, rent until day $\mathsf{b} - 1$ and buy on day $\mathsf{b}$ has a competitive ratio of 2. It is a less trivial fact that there exists *randomized algorithms* with a competitive ratio $\frac{e}{e-1} \approx 1.58198$, see [9, 10, 7, 16, 14].

Suppose the skier has access to a (non-clairvoyant) predictor which predicts the number $\hat{T}$ - an estimate of the true value of $\mathbf{T}$. The goal then is to incorporate this extra information to improve the performance of the algorithm.

**A toy example**  In [15, Algorithm 2] the authors provide a deterministic algorithms that is $(1+\lambda^{-1})$-robust and $(1 + \lambda)$-consistent, where $\lambda \in (0, 1)$. This algorithm *only* depends on the nature of the prediction: if $\hat{T} > \mathsf{b}$ or $\hat{T} < \mathsf{b}$; it doesn't use the prediction itself. In Algorithm 1 below, we consider a minor modification showing how to leverage the information contained in the value of the prediction $\hat{T}$ to improve either robustness or consistency.

---

**Algorithm 1** A modified version of [15, Algorithm 2].

---

**Require:** Parameters $\mathsf{b}, \mathbf{T}, \hat{T}, \lambda \in (0, 1)$.
  Functions $\mu_1(\lambda, \mathsf{b}, \hat{T}), \mu_2(\lambda, \mathsf{b}, \hat{T})$ are described in Proposition 2.1 below.
  **if** $\hat{T} \geq \mathsf{b}$ **then**
    buy on $\mu_1(\lambda, \mathsf{b}, \hat{T})\mathsf{b}$
  **else if** $\hat{T} < \mathsf{b}$ **then**
    buy on $\mu_2(\lambda, \mathsf{b}, \hat{T})\mathsf{b}$
  **end if**

---

**Proposition 2.1.** *(Optimizing robustness) Set,* $\mu_1 = \max\{\lambda, \hat{T}/\mathsf{b}\}$ *when* $\hat{T} > \mathsf{b}$ *and* $\mu_2 = \min\{\mathsf{b}/\lambda, \mathsf{b}/\hat{T}\}$ *when* $\hat{T} < \mathsf{b}$*; then Algorithm 1 is* $\max\{(1 + \mathsf{b}/\mu_1), (1 + \mu_2/\mathsf{b})\}$*-robust.*

*(Optimizing consistency) Set,* $\mu_1 = \min\{\lambda, \hat{T}/\mathsf{b}\}$ *when* $\hat{T} > \mathsf{b}$ *and* $\mu_2 = \min\{\mathsf{b}/\lambda, \mathsf{b}/\hat{T}\}$ *when* $\hat{T} < \mathsf{b}$*; then the algorithm is* $\max\{(1 + \mu_1/\mathsf{b}), (1 + \mu_2/\mathsf{b})\}$*-consistent.*

The proof of this proposition is very similar to the proof of [15, Theorem 2.2] and we omit the details. In §2.3, we compare the empirical performance of Algorithm 1 and Algorithm 2 of [15] on the same data set.

### 2.1 Hedge($\beta$) algorithm for ski rental problem

We now present a *multiplicative weight algorithm* for the ski-rental problem.

For brevity, let 0 denote *rent* and 1 denote *buy*. We consider the three element *decision space*

$$\mathcal{A} = \{0|0, 1|0, 1|1\}.$$

Intuitively, on a given day $t$ the action $0|0$ means renting on day $t$ given that it was rented on day $t - 1$, $1|0$ buying on day $t$ given that it was rented the day before and finally $1|1$ corresponds to (virtually) buying the equipment on day $t$ provided it was already purchased on day $t - 1$ [1]. On the first day, the actions $1|0$ and $1|1$ both lead to buying.

The algorithm proceeds as follows: We choose two parameters $\beta \in (0, 1), \rho > 0$. (The significance of these parameters will be discussed below in §2.1.1.) If it snows on day $t$, then we assign weight

$w_a^t$ to action $a \in \mathcal{A}$. Subsequently, an action $a$ is chosen with probability $p_a^t = w_a^t / \sum_{k \in \mathcal{A}} w_k^t$. The weights are then updated $w_a^{t+1} = \beta^{\ell_a^t/\rho} w_a^t$, where the loss $\ell_a^t$ corresponding to action $a$ is tabulated in Table 1 below.

The algorithm runs for $\mathbf{T}$ days. Complete details are presented in Algorithm 2 below. In hindsight, the optimal actions $A_{\mathsf{opt}}$ are:
(*a*) If $\mathbf{T} \geq$ b then $A_{\mathsf{opt}} = (1|1, 1|1, \ldots) \in \mathcal{A}^{\mathbf{T}}$;
and (*b*) If $\mathbf{T} <$ b then $A_{\mathsf{opt}} = (0|0, 0|0, \ldots) \in \mathcal{A}^{\mathbf{T}}$.

| Action $a \in \mathcal{A}$ | Loss $\ell_a^t$ |
|:---:|:---:|
| 0\|0 | 1 |
| 1\|0 | b |
| 1\|1 | b if $t = 1$, 0 otherwise |

Table 1: Cost of actions.

---

**Algorithm 2** Hedge($\beta$) algorithm for ski rental problem.

---

**Require:** Parameters $\beta \in (0,1)$, $\rho > 0$, $\mathcal{A} = \{0|0, 1|1, 1|0\}$.
 1: Initialize weights $w_a^1 = 1/|\mathcal{A}|$ for $a \in \mathcal{A}$
 2: **for** $t = (1, 2, \ldots, \mathbf{T})$ **do**
 3:     Set $p_a^t \leftarrow w_a^t / \sum_{a \in \mathcal{A}} w_a^t$ for $a \in \mathcal{A}$.
 4:     Select $a \in \mathcal{A}$ with probability $p_a^t$
 5:     **if** $a = 1|0$ or $a = 1|1$ **then** exit                $\triangleright$ Conditional Termination
 6:     **else**
 7:         Update Weights $w_a^{t+1} \leftarrow \beta^{\ell_a^t/\rho} w_a^t$ for $a \in \mathcal{A}$         $\triangleright$ See Table 1 for $\ell_a$.
 8:     **end if**
 9: **end for**

---

We have the following bounds on the competitive ratio of the above algorithm.

**Theorem 2.1.** *Let $\beta \in (0,1)$ and $\rho > 0$. Then, we have an upper bound on the expected cost of the algorithm $\mathsf{E}(\mathcal{C}_{\mathsf{alg}})$ in terms of the optimal cost $\mathcal{C}_{\mathsf{opt}}$. It is given by the equation*

$$\mathsf{E}(\mathcal{C}_{\mathsf{alg}}) \leq \frac{\ln(\beta^{1/\rho})}{\beta^{1/\rho} - 1} \mathcal{C}_{\mathsf{opt}} + \frac{\ln(2)}{\beta^{1/\rho} - 1}.$$

*Proof.* The competitive ratio of Algorithm 2 *without* the termination condition (see line 5) can only become larger. We analyze the expected cost without this condition.

Consider the potential $\Phi^t = \sum_{a \in \mathcal{A}} w_a^t$. We have the inequality

$$\Phi^{t+1} = \sum_{a \in \mathcal{A}} \beta^{\ell_a^t/\rho} w_a^t \leq \left( \sum_a (1 - (1 - \beta^{1/\rho}) \ell_a^t) p_a^t \right) \Phi^t = (1 - (1 - \beta^{1/\rho}) \sum_a p_a^t \ell_a^t) \Phi^t. \quad (1)$$

At $t = \mathbf{T}$, we get $\Phi^{\mathbf{T}} \leq \prod_{t=1}^{\mathbf{T}} (1 - (1 - \beta^{1/\rho}) \sum_a p_a^t \ell_a^t) \leq e^{-(1 - \beta^{1/\rho}) \sum_{t=1}^{\mathbf{T}} \sum_a p_a^t \ell_a^t}$. We note that $\sum_{t=1}^{\mathbf{T}} \sum_a p_a^t \ell_a^t = \mathsf{E}(\mathcal{C}_{\mathsf{alg}})$, so $\Phi^{\mathbf{T}} \leq e^{-(1 - \beta^{1/\rho}) \mathsf{E}(\mathcal{C}_{\mathsf{alg}})}$. On the other hand, for *any* fixed action $a \in \mathcal{A}$, we have $w_a^1 \beta^{\mathbf{T} \ell_a/\rho} = w_a^1 \beta^{\mathcal{C}_{\mathsf{opt}}/\rho}$. Combining the upper and lower bounds on $\Phi^{\mathbf{T}}$ and using $w_a^1 \leq 1/2$ we get the desired bound. $\square$

### 2.1.1 The parameters $\rho$ and $\beta$

The parameter $\beta$ is central to the algorithm. It is important because at $\beta = 0.4$ the value of $\frac{\ln(\beta)}{\beta - 1} < \frac{e}{e-1}$.

The parameter $\rho$ acts as a *normalizing factor* that is computationally useful in implementing Algorithm 2. It allows us to avoid underflow errors. It has several different interpretations in the multiplicative weight context, see [1]. In the original treatment of Hedge($\beta$) algorithm $\rho$ is set to 1.

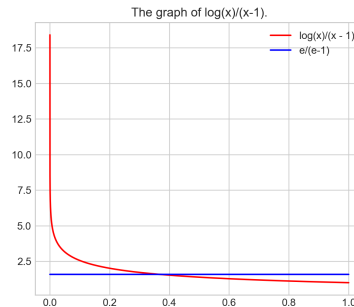

## 2.2 Hedge($\beta$) with ML prediction

An ML predictor provides us with an estimate $\hat{T}$ of the true but unknown value of $\mathbf{T}$. We say that a ML advisor is $\alpha$-accurate if the conditional probabilities

$$\Pr[\mathbf{T} \geq \mathsf{b}|\hat{T} \geq \mathsf{b}] \geq \alpha \implies \Pr[\mathbf{T} < \mathsf{b}|\hat{T} \geq \mathsf{b}] < 1 - \alpha$$

$$\text{and} \Pr[\mathbf{T} < \mathsf{b}|\hat{T} < \mathsf{b}] \geq \alpha \implies \Pr[\mathbf{T} \geq \mathsf{b}|\hat{T} < \mathsf{b}] < 1 - \alpha$$

where $\alpha \in [1/2, 1]$. A value of $\alpha = 1$ corresponds to a perfect predictor whereas $\alpha = 1/2$ corresponds to random uninformative predictor.

Intuitively, we incorporate the ML-prediction and its accuracy into the Hedge($\beta$) as a prior. Given a prediction, say $\hat{T} < \mathsf{b}$, imagine that we have $\mathbf{T}$ (which is unknown) advisors. On the first day, the proportion $\alpha$ of the advisors suggest renting and the rest suggest buying. We assume that among these $\alpha \mathbf{T}$-advisors, who suggest renting, the prediction that $\mathbf{T} = i$, for $1 \leq i \leq \mathsf{b} - 1$, are equally likely [2]. So, on day $i$ we discard the incorrect predictions i.e. we remove $(i - 1)\alpha \mathbf{T}/\mathsf{b}$ advisors and after adjusting the prior accordingly (see line 12 in Algorithm 3 below) we decide between renting or buying.

**Theorem 2.2.** *Let $\beta \in (0, 1)$ and $\rho > 0$. Then, we have an upper bound on the expected cost of the Algorithm 3, $\mathsf{E}(\mathcal{C}_{\mathsf{alg}})$, in terms of the optimal cost $\mathcal{C}_{\mathsf{opt}}$ is given by*

$$\mathsf{E}(\mathcal{C}_{\mathsf{alg}}) \leq \frac{\ln(\beta^{1/\rho})}{\beta^{1/\rho} - 1}\mathcal{C}_{\mathsf{opt}} + \frac{\ln(\alpha)}{\beta^{1/\rho} - 1}. \tag{2}$$

The fundamental inequality, equation 1, used in the proof of the Theorem 2.1, also holds for Algorithm 3. As a result, the proof of the above proposition follows from the same reasoning.

*Remark* 2.1. We do not know of a closed analytic formula that explicates the dependence of the competitive ratio on $\alpha$. We empirically observe the better performance of this algorithm in simulations, see §2.3.

When $\mathbf{T} \gg \mathsf{b}$, Algorithm 3 has the added benefit that is terminates after $\mathsf{b}$ iterations.

---

**Algorithm 3** Hedge($\beta$) algorithm with an $\alpha$-accurate ML advisor.

---

**Require:** Parameters $\beta \in (0, 1)$, $\rho > 0$, $\mathcal{A} = \{0, 1\}$. ML-Prediction $\hat{T}$ and its accuracy $\alpha \in [1/2, 1]$.
1: **if** $\hat{T} \geq \mathsf{b}$ **then**
2:      Initialize prior $\pi_0 = 1 - \alpha, \pi_1 = \alpha$
3: **else** $\hat{T} < \mathsf{b}$
4:      Initialize prior $\pi_0 = \alpha, \pi_1 = 1 - \alpha$
5: **end if**
6: Initialize weights $w_0^1 = \pi_0, w_1^1 = \pi_1$.
7: **for** $t = (1, 2, \ldots, \min(\mathbf{T}, \mathsf{b}))$ **do**
8:      $p_a^t \leftarrow w_a^t / \sum_{a \in \mathcal{A}} w_a^t$ for $a \in \mathcal{A}$.
9:      Select $a \in \mathcal{A}$ with probability $p_a^t$
10:      **if** $a = 1$ **then** exit                                      ▷ Conditional Termination
11:      **else**
12:          Update prior: $\pi_0 \leftarrow \frac{\pi_0 - \pi_0/\mathsf{b}}{1 - \pi_0/\mathsf{b}}$ and $\pi_1 \leftarrow \frac{\pi_1}{1 - \pi_0/\mathsf{b}}$
13:          Update weights: $w_a^{t+1} \leftarrow \pi_a \beta^{\ell_a^t/\rho} w_a^t$ for $a \in \mathcal{A}$          ▷ Where $\ell_0 = 1, \ell_1 = \mathsf{b}$.
14:      **end if**
15: **end for**

---

## 2.3 Experiments

We study the empirical performance of our algorithms for the ski rental problem via simulations. We set $\mathsf{b} = 100$ and the number of snow days $\mathbf{T}$ is drawn uniformly from $[1, 4\mathsf{b}]$.

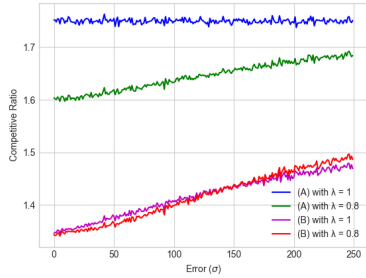
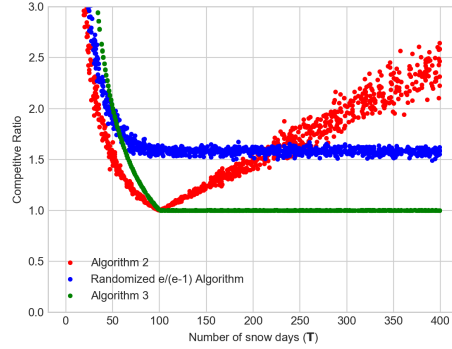

Figure 1: (A) = Algorithm 2 of [15] & (B) = Algorithm 1. Empirical performance on synthetic data.

Figure 2: We consider the competitive ratios of Algorithm 2 in red, the randomized $\frac{e}{e-1}$ algorithm in blue and Algorithm 3 in green.

In Figure 1, following the data generation scheme of [15], we generate predictions $\hat{T} = \min(\mathbf{T} + \epsilon, 1)$ where $\epsilon$ is drawn from a normal distribution with mean 0 and variance $\sigma$. For each value of $\sigma$ we plot the average competitive ratio of 10000 independent trials. The blue (●) and green (●) plots show the average competitive ratios of Algorithm 2 of [15] at $\lambda = 1, 0.8$ respectively. The red (●) and the magenta (●) plots show the average competitive ratio of Algorithm 1 with $\mu_1(x, y) = \mu_2(x, y) = \min(x, y)$ and $\lambda = 1, 0.8$ respectively. Figure 1 shows that using the information in the prediction can lead to significant improvement in the competitive ratio.

In Figure 2, we generate 1000 samples of $\mathbf{T}$ uniformly from $[1, 4b]$. We plot the average competitive ratio of 100 independent trials with $\hat{T}$ generated with accuracy $\alpha = 0.8$. We fix $\beta$ at 0.5 and $\rho$ at 20 respectively. Algorithm 2 is plotted in red (●) and Algorithm 3 is plotted in green (●). The blue (●) plot presents average compete ratio (over the same data) for the classical randomized $e/(e-1)$ algorithm.

In Figure 2, we observe that when $\mathbf{T}/b < 0.5$ the constant overhead (the term $\ln(\alpha)/(\beta^{1/\rho} - 1)$ in Equation 2) is dominant. When $\mathbf{T}/b \geq 1$, Algorithm 3 outperforms Algorithm 2 and the randomized $e/(e-1)$ algorithm. This is expected because when $t \gg 0$ the the weights $w_a^t$ in Algorithm 2 can suffer from large underflow errors whereas underflow error in Algorithm 3 is bounded.

## 3 Dynamic TCP acknowledgment problem

Data packets arrive at a location at times $0 < a_1 < \ldots < a_n < \infty$. We call $\mathcal{A} = (a_1, a_2, \ldots, a_n)$ an *incoming sequence*. All incoming data packets must be *acknowledged* at some point of time after their arrival. It is possible to acknowledge multiple incoming packets together. The cost of a single acknowledgment is 1.

At time $t$, the *latency* of a subset $S \subset \mathcal{A}$ is given by

$$\mathsf{Lat}(S, t) = \sum_{a_j \in S} (t - a_j) 1_{\{a_j < t\}},$$

where $1_{\{a_j < t\}}$ is the indicator function. (We disregard any data packets in $S$ after time $t$.)

A *schedule* $\pi$ is a partition of an incoming sequence $\mathcal{A}$ into disjoint contiguous subsets. A *feasible* acknowledgment time, adapted to a schedule $\pi$ is a monotonically increasing ordered set $\mathcal{T} = (t_J : J \in \pi)$ such that for every $J \in \pi$ we have $t_J \geq \max_{a_i \in J} a_i$. The *cost* of a schedule $\pi$ with an acknowledgment time $\mathcal{T}$ is given by

$$\mathsf{Cost}(\pi, \mathcal{T}) = |\pi| + \sum_{J \in \pi} \mathsf{Lat}(J, t_J). \tag{3}$$

In the offline setting, when $\mathcal{A}$ is known and the pair $(\pi_{\mathsf{opt}}, \mathcal{T}_{\mathsf{opt}})$ minimizes the total cost among all schedules and feasible acknowledgment times, i.e.

$$\mathsf{Cost}(\pi_{\mathsf{opt}}, \mathcal{T}_{\mathsf{opt}}) = \min_{(\pi, \mathcal{T})} \mathsf{Cost}(\pi, \mathcal{T}).$$

In [4], Dooley et al. describe an offline algorithm that produces $(\pi_{\mathsf{opt}}, \mathcal{T}_{\mathsf{opt}})$ for any incoming sequence $\mathcal{A}$.

In the online setting randomized algorithms with competitive ratio $\frac{e}{e-1}$ is known due to the work of several authors, see [9, 16, 14]. We recall Seiden's optimal offline algorithm below.

---

**Algorithm 4** Seiden's sequential algorithm for the Dynamic Online TCP problem

---

**Require:** Let $P(z)$ denote the continuous probability distribution on $[0, 1]$ with density $\frac{e^z}{e-1}$.

1: **loop**
2:     Pick $x$ from $P(z)$.
3:     **if** no more packets are coming **then**
4:         If needed, send the final acknowledgment and exit.
5:     **else if** no more packets arrive before time $x$ **then**
6:         If needed send acknowledgment for all outstanding packets at time $x$.
7:     **end if**
8: **end loop**

---

## 3.1   ML predictor

The output from a ML-predictor provides a schedule $\pi^{ML}$ and a feasible acknowledgment time $\mathcal{T}^{ML}$ (adapted to $\pi^{ML}$). Now for an incoming sequence $\mathcal{A}$ it may happen that either $\pi^{ML}$ is not a schedule for $\mathcal{A}$ (it is not a complete partition) or $\mathcal{T}^{ML}$ is not feasible. In either case, we can canonically *enlarge* $\pi^{ML}$ by appending any outstanding packets and adding a final acknowledgment at the very end to $\mathcal{T}^{ML}$. After this necessary extension[3], in general $(\pi^{ML}, \mathcal{T}^{ML})$ will not be optimal for $\mathcal{A}$ anymore.

**Definition 3.1.** In the TCP setting, we say that a ML advisor is $\epsilon$-close and $\alpha$-accurate if for all input sequences $\mathcal{A}$ the tail probability

$$\Pr[\mathsf{Cost}(\pi^{ML}, \mathcal{T}^{ML}) > (1 + \epsilon)\,\mathsf{Cost}_{\mathsf{opt}}] < 1 - \alpha.$$

Here $\alpha \in [0, 1]$ and it implicitly depends on $\epsilon$.

Since $\frac{e}{e-1}$ randomized algorithms are known, in practice one should expect $\epsilon < 1/(e-1)$ and the corresponding $\alpha$ is close to one. A simple consequence of the above definition that holds for any reasonable $(\epsilon, \alpha)$ ML-predictor is that

$$\mathsf{E}[\mathsf{Cost}(\pi^{ML}, \mathcal{T}^{ML})] \leq \left(\frac{e}{e-1}(1 - \alpha) + (1 + \epsilon)\alpha\right)\mathsf{Cost}(\pi_{\mathsf{opt}}, \mathcal{T}_{\mathsf{opt}}). \quad (4)$$

We present a Hedge($\beta$) online algorithm with an ML-predictor with accuracy $(\epsilon, \alpha)$, see Algorithm 5. The algorithm performs well with accurate predictions while hedges against less accurate predictions.

Denoting the expected cost of Algorithm 5 by $\mathsf{E}(\mathcal{C}_{MW})$ and that of the ML advisor by $\mathsf{E}(\mathcal{C}_{ML})$ we have

$$\mathsf{E}(\mathcal{C}_{MW}) \leq \frac{-\log(\beta)\,\mathsf{E}(\mathcal{C}_{ML}) - \log(\alpha)}{1 - \beta}.$$

Combining the above bound with the equation 4 we notice that a ML advisor with parameters $(\epsilon, \alpha)$ improves the online TCP acknowledgment algorithm if

$$\frac{\log(\beta)}{\beta - 1}\left(\frac{e}{e-1}(1 - \alpha) + (1 + \epsilon)\alpha\right) < \frac{e}{e-1}.$$

**Algorithm 5** A Hedge($\beta$) algorithm from the Dynamic TCP problem. We assume we are provided with a ML advisor with parameters ($\epsilon, \alpha$).

---

**Require:** A ($\epsilon, \alpha$) ML advisor providing acknowledgment sequence $\mathcal{T}^{ML} = (t_1^{ML}, \ldots, t_k^{ML})$ and the probability distribution $P(z)$ with density $e^z/(e-1)$.

1: Set weights, $w_0 = 1 - \alpha$, $w_1 = \alpha$.
2: **loop**
3:　　Pick $i \in \{0, 1\}$ with probability $p_i = w_i/(w_0 + w_1)$.
4:　　**if** $i = 0$ **then**
5:　　　　Pick $x$ from $P(z)$.
6:　　　　$t \leftarrow x$
7:　　**else**
8:　　　　$y \leftarrow$ the next element of $\mathcal{T}^{ML}$
9:　　　　$t \leftarrow y$
10:　　**end if**
11:　　**if** no more packets are coming **then**
12:　　　　If needed, send the final acknowledgment and exit.
13:　　**else if** no more packets arrive before time $t$ **then**
14:　　　　$S \leftarrow$ arrival time of all the outstanding packets acknowledged at time $t$
15:　　　　Send acknowledgment
16:　　　　Update weights $w_0 \leftarrow \beta^{1+\mathsf{Lat}(S,x)} w_0$ and $w_1 \leftarrow \beta^{1+\mathsf{Lat}(S,y)} w_1$
17:　　**end if**
18: **end loop**

---

In particular, for an ML setup to improve competitive ration it is necessary that its accuracy $\alpha$ satisfy

$$(1 - \alpha) \frac{\log(\beta)}{\beta - 1} < 1.$$

## 4   Conclusion and Further Work

In this paper we show that one can use the multiplicative weight update method to boost online rent/buy problems with ML predictions in a natural way. It will be interesting to see if it is possible to extend such a framework to other online problems.

Practical performance of online algorithms have been studied, see [8]. It will be interesting to construct practical ML-systems that can learn and accurately predict distribution of incoming inputs. The results of this paper suggest that beyond a certain threshold can provide significant performance gains.

### Broader Impact

The current paper presents theoretical results that revisits well known problems in computer science using established ideas from machine learning. Our work presents a transparent framework where online algorithms and predictions can be merged seamlessly to improve algorithmic performance. This can help with cost benefit analysis of deploying ML systems to boost performance in processes that require algorithmic decision making in the face of uncertain inputs.

We do not see any conceivable way in which this work will put any section of society at a disadvantage or amplify biases in input data.

### Acknowledgments and Disclosure of Funding

The author gratefully acknowledges partial financial support from Minnesota State University (MNSU) in summer 2020 that helped complete the current work.

## Footnotes

[1]The decision $1|1$ acts as a useful placeholder.

[2] One can use a different distribution if more information is available.

[3]For brevity, we use the same notation – $\pi^{ML}$ and $\mathcal{T}^{ML}$ – to denote these extensions.

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
