[Reviews · NeurIPS 2020]

Review 1

Summary and Contributions: Comments after rebuttal: thanks to the authors for addressing some of the issues raised in my review. It is a pity though that the authors did not implement the e/(e-1)-competitive randomized algorithm, it is a really simple algorithm. Overall, my score remains unchanged. ---- The paper considers online algorithms that make use of predictions. Such algorithms provide worst-case guarantees, while utilizing predictions to improve their performance if those predictions are close to correct. Two specific problems considered are ski rental and TCP acknowledgment. Both are classic problems well-studied in the online algorithm literature. The paper presents algorithms for the aforementioned problems, as well as their experimental evaluation (on synthetic data).

Strengths: - The algorithms that incorporate multiplicative weights approach. This seems like an interesting research direction. - For some range of parameters, the proposed algorithms improve (in theory and experimentally) over prior bounds

Weaknesses: - It is not clear how natural the synthetic data sets are, and how they were selected. For example, ski rental is evaluated on instances selected uniformly at random from the intervals {1....4b} or {1...2.5b}. What is the reason for choosing these ranges ? Why are they different ? Note that if the instances were instead selected uniformly at random from {1...b}, then the classic 2-competitive algorithm would actually achieve optimal cost, so the proposed algorithms would be not better (and probably worse) than what is known. So the size of the range matters. - It would also be helpful to experimentally compare the proposed algorithms to the classic approaches (2-competitive deterministic and e/(e-1)-competitive randomized).

Correctness: Yes, although see comments about experiments above.

Clarity: Yes.

Relation to Prior Work: Yes.

Reproducibility: Yes

Additional Feedback:


Review 2

Summary and Contributions: The paper considers ski rental and related online problems, in the context of predictions, primarily based on using the Hedge algorithm.

Strengths: This provides a nice framework, extending some of the ideas of consistency and robustness in previous work, providing potentially useful frameworks for further work.

Weaknesses: Most everything seems to boil down to "Use a standard hedge algorithm along with the prediction as a prior", which isn't especially compelling. The experiments seem poorly designed.

Correctness: The proofs and such are correct. The authors need a more robust set of experiments.

Clarity: The paper is a bit notation heavy at times. I consider labeling the 3 decisions for ski rental 0|0, 1|0, 1|1, for example, annoying and unhelpful.

Relation to Prior Work: Relation to prior work is discussed.

Reproducibility: Yes

Additional Feedback: There were many thinks liked about the paper, including the idea of having an ML-advisor algorithm be e-close and alpha-accurate. I also liked the interpretation of a Hedge algorithm with an advisor give on page 5. In some ways the Hedge formalization, though, seems to minimize the use of predictors to just giving a prior. As someone interested in this area I find that a not especially hopeful or compelling message, but perhaps for some problems that's the right methodology. The experiments don't seem that useful. In particular, for Figure 1, choosing lambda = 1 just gives the "standard" 2-competitive type algorithm for the ski rental problem, so the results are unsurprising. (Ideally, one should vary lambda with sigma, as ostensibly one has some idea of the noise.) Indeed, using normally distributed noise for the predictor is somewhat disappointing and does not, I think, show the strength or weaknesses of predictors adequately. I realize you are short on space, but better experiments (with other failure modes in the predictors) seems important to test. Some discussion of Figure 2 seems warranted. The results look interesting -- what should we be taking away from the Figure? Post feedback: Overall I thought the authors responded well to the review concerns. While I am not changing my score, I remain favorably disposed to acceptance.


Review 3

Summary and Contributions: The authors studied the online rent-or-buy problem and how predictions made by machine learning can be incorporated. The authors proposed a new algorithm where performance guarantees come in terms of the accuracy of the prediction, where accuracy can be problem-specific. The algorithmic framework is applied to both the ski-rental problem and the dynamic TCP acknowledgment problem and is shown to be closed to optimal with accurate predictions via simulation.

Strengths: The overall idea of using ML prediction to improve performance for online algorithm when prediction error is small while still keeping the worst case guarantee is appealing. However, this seems to come from [13].

Weaknesses: 1. There is not much distinction in terms of novelty in methodology and theoretical results compared to [13]. The authors introduced the accuracy measure for prediction uncertainty but did not fully justify why this is a good measure. 2. The theoretical results are a little weak, the accuracy term does not improve the competitive ratio in Theorem 2.2, rather, it only appears in the constant term. 3. The robustness and consistency guarantee of Algorithm 1 come from two different set of parameter setting, this seems a little strange. Is it possible to have both robustness and consistency guarantee with a common set of parameters? 4. It would be good to show the actual performance of Algorithm 5 compared to existing online algorithm

Correctness: The claims and methods seem to be correct.

Clarity: Overall the presentation is clear.

Relation to Prior Work: The distinction between this work and [13] is not very clear.

Reproducibility: Yes

Additional Feedback:

[Author Response · NeurIPS 2020]

We thank the reviewers for their feedback. Additional feedback and suggestions for improvement from Reviewer #3 is
greatly appreciated. We address the issues and concerns raised by the reviewers below.

*The data-sets used in experiments [R1 & R3].* The treatment of the ski-rental problem with ML-advisor in [1] is germane
to our present work. In Fig. 1 we use the same simulation setup as that of [1] (cf. Pg. 8 §4.1 of [1]) to provide a
transparent comparison of algorithmic performance. We strongly agree that using non-normal noise is important, but
such a discussion within the limited space will significantly deviate from the main idea of the present paper.

The depiction of the range of $\mathbf{T}$ in Fig. 2 was driven by space constraints.
Simulating Algorithms 2 and 3 on the range $[1, 4\mathbf{b}]$ instead of $[1, 2.5\mathbf{b}]$
did not provide radically new insights. The graph on the right presents
Algorithm 2 (in red) and Algorithm 3 (in green) with $\mathbf{T} \in [1, 4\mathbf{b}]$.

*Comparison with classic approaches [R1].* Some discussion on the
comparison between classic approaches and the recent ones is presented
in [1]. The performance of the classic randomized $e(e-1)^{-1}$-algorithm
in the experiments leading to Fig. 2 will be included in the updated
version.

*Takeaways from Fig. 2 [R3].* (1) In algorithmic implementation, the hyperparameter $\rho$ plays a significant role. We fixed
$\rho = 20$ in experiments. This lead to underflow issues when $\mathbf{T} \gg \mathbf{b}$, as demonstrated by the increasing trend in the red
ticks in Fig. 2. This suggests, one should vary $\rho$ within an appropriate range. (2) Algorithm 3 avoids this issue at the
expense of performance when $\mathbf{T} \ll \mathbf{b}$ (higher values of green ticks in that range). (3) The relatively high competitive
ratio of the hedge algorithm, when $\mathbf{T} \ll \mathbf{b}$, corresponds to the to the fixed overhead constant term in Theorem 2.1.

*The efficacy of the accuracy measure introduced in this paper [R4].* In statistical prediction, a biased estimator with
low variance can be more effective than a consistent estimator with unknown variance. Hypothetically, consider two
predictors A and B for ski-rental problem. Suppose, for simplicity, A has a MSE of 3 days and B has a MSE of 5
days. The consistency/robustness framework does not justify why A is a better choice than B. Our accuracy measure
provides a justification. A quantitative discussion is presented in lines $44 - 51$ of the paper. Moreover, it is unclear how
to extend robustness/consistency to the TCP problem where the decision space is $\Delta = [0, 1]^n$. Such an extension will
be sensitive to the metric used on $\Delta$ (implicit in the definition of $\eta$). The $(\epsilon, \alpha)$-accuracy measure extends in this setting
with relative ease and conceptual clarity.

*The accuracy term doesn't improve the competitve ratio in Theorem 2.2 [R4].* Ensuring that the competitive ratio
doesn't inflate significantly due to bad predictions is a central design feature. This fails if the competitive ratio has a
direct functional dependence on accuracy. However, Thm 2.2 demonstrates how to hedge against bad predictions while
leveraging accurate predictions (alpha = 1) – thus retaining the essence of online algorithms.

*The robustness and consistency guarantee of Algorithm 1 coming from two different set of parameter settings. [R4]* The
guarantees come from two different functions (not parameters) which depend on *a single* parameter $\lambda$ and the data.
The takeaway from Proposition 2.1 and Fig. 1 of the present paper is that using a hyperparameter and the *information*
conveyed by the predictor ($\hat{\mathbf{T}}$) significantly outperforms an algorithm just using the hyperparameter alone (cf. Theorem
2.2 and Fig. 2(a) of [1]). The same numerical value of the hyperparameter never optimizes robustness and consistency
simultaneously (cf. the $(1 + \lambda^{-1})$-robust and $(1 + \lambda)$-consistent Algorithm 2 of [1]). This is unsurprising because an
optimal robust algorithm must also hedge against bad predictions while an optimal consistent algorithm assumes perfect
prediction (cf. line 42 of the paper).

*Comparing Algorithm 5 to an existing online algorithm [R4].* We are not aware of any prior work on Dynamic TCP
Acknowledgment problem with ML prediction. As observed in line 197, Algorithm 5 can improve over the classic
$e(e-1)^{-1}$-randomized algorithm only under strict accuracy guarantees from the ML-predictor.

*The distinction between this work and the methodology and theoretical results in [1] [R4].* Previous work relevant to
the current paper is discussed in lines $31 - 32$. In this paper, we present a new accuracy measure and compare it with
existing framework of [1] (cf. lines $40 - 56$). Algorithm 1 presents a modification of Algorithm 2 of [1]. With the
goal of providing a clear and transparent comparison we use the same simulation framework to compare the relative
performances of these algorithms. The rest of the present paper is about other algorithms which use a completely
different approach from that of [1]. Additionally, we discuss the Dynamic TCP acknowledgment problem within our
framework. This problem has not been considered in [1].

# References

[1] Purohit, M., Svitkina, Z., and Kumar, R. *Improving online algorithms via ML predictions*, Advances in Neural Information
Processing Systems, 9661–9670, 2018.


[Meta-Review · NeurIPS 2020]

The paper studies online rent-or-buy problems (ski-rental problem and the dynamic TCP problem) as a sequential decision making problem. It is shown how one can integrate predictions, typically coming from a machine learning algorithm, into this framework using a multiplicative weight algorithm. The paper has been positively evaluated by all the reviewers, with a uniform score of 6. The reviewers liked the overall idea of using ML predictions to improve the performance of online algorithms while still keeping the worst case guarantees, as well as incorporation of the multiplicative weights algorithm. On the other hand, the novelty of the paper seems a bit weak from a methodological perspective: the authors apply a well-known Hedge algorithm with ML predictions just incorporated within the prior. Also, contribution comparing to [13] seems somewhat incremental. Finally, the computational experiments do not seem clear and useful.